# To Trim or Not to Trim: Effects of Read Trimming on the De Novo Genome Assembly of a Widespread East Asian Passerine, the Rufous-Capped Babbler (*Cyanoderma ruficeps* Blyth)

**DOI:** 10.3390/genes10100737

**Published:** 2019-09-23

**Authors:** Shang-Fang Yang, Chia-Wei Lu, Cheng-Te Yao, Chih-Ming Hung

**Affiliations:** 1Biodiversity Research Center, Academia Sinica, Taipei 11529, Taiwan; frankswatch@msn.com (S.-F.Y.);; 2Division of Zoology, Endemic Species Research Institute, Nantou 552, Taiwan; yaoct@tesri.gov.tw

**Keywords:** de novo genome assemble, reading trimming, rufous-capped babbler, computational time, genome quality

## Abstract

Trimming low quality bases from sequencing reads is considered as routine procedure for genome assembly; however, we know little about its pros and cons. Here, we used empirical data to examine how read trimming affects assembled genome quality and computational time for a widespread East Asian passerine, the rufous-capped babbler (*Cyanoderma ruficeps* Blyth). We found that scaffolds assembled from raw reads were always longer than those from trimmed ones, whereas computational times for the former were sometimes much longer than the latter. Nevertheless, assembly completeness showed little difference among the trimming strategies. One should determine the optimal trimming strategy based on what the assembled genome will be used for. For example, to identify single nucleotide polymorphisms (SNPs) associated with phenotypic evolution, applying PLATANUS to gently trim reads would yield a reference genome with a slightly shorter scaffold length (N50 = 15.64 vs. 16.89 Mb) than the raw reads, but would save 75% of computational time. We also found that chromosomes Z, W, and 4A of the rufous-capped babbler were poorly assembled, likely due to a recently fused, neo-sex chromosome. The rufous-capped babbler genome with long scaffolds and quality gene annotation can provide a good system to study avian ecological adaptation in East Asia.

## 1. Introduction

Genomic analyses such as comparative genomics, population genomics, and transcriptomics have advanced our knowledge on diverse biological fields [1]. For example, genome-wide data have helped identify the genetic bases of morphological change or ecological adaptation for organisms in nature [2,3]. Such research may require reference genomes that are often unavailable for non-model species. Thus, the demands for de novo assembling genomes from short sequence reads have been increasing in many biological fields. However, sequence reads do not always have correct signals at all their bases due to errors in sequencing procedures or base calling [4]; low quality bases might prevent correct genome assembly. Removing low quality bases from sequence reads (i.e., read trimming) is often assumed as necessary for de novo genome assembly [5]. Therefore, one usually executes the trimming step without evaluating its costs and benefits on genome assembly.

Different trimming strategies might lead to different genome assembly results and thus should be carefully examined. Trimming low quality bases may not only lower base error rates, but also reduce read numbers and lengths, which could increase (or decrease) assembled genome contiguity and completeness [6,7]. In addition, read trimming could potentially reduce computational time [6]. As sequencing costs have decreased dramatically over the past decade [8,9], budgeting time for genomic analyses has become an increasingly important factor, especially for research teams with limited computational resources. Evaluation based on empirical data to explore different trimming strategies is often needed as the assembled genome quality and computation time may vary among taxa and assembly approaches. In addition, researchers should take into account the applications of assembled genomes to determine the optimal trimming strategy for genome assembly.

The rufous-capped babbler (*Cyanoderma ruficeps* Blyth) is a widespread passerine in East Asia [10]. This bird occurs in diverse bioclimatic zones and thus provides a good system to examine the genetic mechanism underlying avian adaptation to different climatic niches. Comparing genomic sequences between populations from different bioclimatic zones will facilitate the identification of candidate genes for local adaptation, and de novo assembling a reference genome is the first step to this end. In this study, we used the rufous-capped babbler as an empirical example to evaluate how different trimming strategies affect de novo genome assembly. Two assembly approaches were used to examine the effects of three trimming strategies on assembled genome quality and computational time. We used assembly summary statistics, gene prediction results, and chromosome-level genome alignments to the chicken (*Gallus gallus* L.), zebra finch (*Taeniopygia guttata* Vieillot), and collared flycatcher (*Ficedula albicollis* Temminck) to access the quality of the assembled genomes. We also discussed the trade-offs among the trimming strategies based on the applications of the assembled genomes. We expect that the rufous-capped babbler genome will offer a good research system to study the ecological adaptation of passerines in East Asia.

## 2. Materials and Methods 

### 2.1. Ethical Guidelines for the Use of Animals

The use of animals in this study was reviewed and approved by the Institutional Animal Care and Use Committee of Academia Sinica (Protocol ID: 16-01-931).

### 2.2. Genome Sequencing

Library preparation and genomic sequencing were conducted at the NGS High Throughput Genomic Core at the Biodiversity Research Center, Academia Sinica (Taipei, Taiwan). A female rufous-capped babbler was chosen for genomic sequencing as female birds have both Z and W chromosomes. The genomic DNA of this bird was sequenced based on one short (paired-end; 2 × 250 bp) library with an expected fragment length of 470 bp and six long (mate-pair; 2 × 150 bp) libraries with expected fragment lengths of 1, 3, 5, 8, 12, and 20 Kb using the Illumina HiSeq 2500 platform. We used FASTQC version 0.11.5 [11] to check the quality of 250 bp paired-end reads. As the quality of the paired-end reads declined severely after the 190th bp, we used TRIMMOMATIC version 0.36 [12] to remove the low-quality bases in two ways. First, we used the default setting (ILLUMINACLIP:TruSeq3_all.fa:2:30:10 SLIDINGWINDOW: 4:15) to trim the paired-end reads. Second, we cut off the last 60 bases (CROP:190) to retain only the first 190 bases of the reads. We also used raw reads for genome assembly to compare the effects of the trimming practices. Therefore, three datasets of paired-end reads, raw PE, trimmed PE, and cut off PE, were used for genome assembly.

We also removed low-quality bases from 150 bp mate-pair reads by NEXTCLIP version 1.3.2 [13] using the settings -*m* 50 (minimum usable read length set to 50 bases) and -*n* 30,000,000 (approximate number of reads allowed: 30 million). Hence, there was only one dataset of mate-pair reads in this study.

The sequence reads were screened for contamination using KRAKEN version 1.0 [14] based on a customized database. The database included (1) the genomes or nucleotides of taxa (i.e., human, archaea, bacteria, plasmid, and viral) from the standard KRAKEN database, and (2) those of seven taxa, lizard, frog, plant, yeast, fish, moth, and fly (represented by *Anolis carolinensis* Voigt, *Xenopus laevis *Daudin, *Arabidopsis thaliana* L. Heynh, *Saccharomyces cerevisiae* Meyen ex E.C. Hansen, *Danio rerio* Hamilton, *Bombyx mori* Linnaeus, and *Drosophila melanogaster* Meigen, respectively) that could contaminate our samples during the sample preparation process (i.e., sample collection, nucleic acid extraction, and library preparation). We downloaded the genomes of the seven additional species from NCBI. To avoid overestimating contamination, we extracted reads identified as human and lizard (the two most closely related taxa to birds in the KRAKEN database) contaminants by KRAKEN, and mapped them back onto a chicken genome using BLAST + version 2.5.0 [15] for further validation. We treated the mapped reads as reads from the rufous-capped babbler (i.e., clean reads) and included them for downstream analyses. As the quality of paired-end reads dropped severely after the 190th bp and thus might impact contamination identification, we only used the cut off PE dataset to identify contaminated reads. We also removed the contaminated reads from the raw PE and trimmed PE datasets according to the above results. We applied the same decontamination procedures to the trimmed mate-pair reads.

### 2.3. Genome Assembly

The rufous-capped babbler genome was assembled using two approaches. (1) PLATANUS version 1.2.4 [16] was used to assemble contigs based on the three datasets of paired-end reads (i.e., raw PE, trimmed PE, and cut off PE), separately. We then constructed scaffolds based on the contigs and trimmed mate-pair reads using the same program. (2) DISCOVARdenovo v52488 [17] was used to assemble contigs based on each of the three paired-end read datasets using default parameters. We used SOAPdenovo2 version 2.04 [18] for scaffolding based on the contigs and trimmed mate-pair reads. We tested a series of kmer lengths ranging from 25 to 121 bp for scaffolding, and selected the 25-mer in this step to generate scaffolds with the longest N50 value. To close gaps in the assembled scaffolds generated from both of the above two approaches, we used GAPCLOSER version 1.12 distributed with SOAPdenovo2. We then used SEQTK [19] to exclude scaffolds shorter than 1000 bp from the downstream analyses including gene prediction and assembly completeness assessment. We referred to genomes assembled using the first approach as PLATANUS-assembled genomes and those using the second approach as DISCOVARdenovo+SOAPdenovo2-assembled genomes. The trimmed PE PLATANUS-assembled genome and raw sequence reads were deposited to the National Center for Biotechnology (NCBI) under the BioProject accession number PRJNA573563.

### 2.4. Gene Prediction for the Assembled Genome

Gene prediction was performed for the three PLATANUS-assembled genomes (which had better assembled qualities than the DISCOVARdenovo + SOAPdenovo2-assembled genomes; see Results for details) using the BRAKER1 pipeline version 1.9 [20]. Before performing gene prediction, we used BUSCO version 3.0.2 [21] to generate training files based on 4915 avian expected (AVES) genes from the BUSCO database. To obtain RNA evidence for gene prediction, we conducted RNA sequencing (RNA-seq) using Illumina HiSeq 2500 on the RNA samples extracted from multiple tissues (i.e., brain, liver, heart, kidney, pancreas, ovary, lung, and pectoral muscle) of the same bird used for genome sequencing. We used TRIMMOMATIC to trim Illumina adapters and low-quality bases from raw RNA (cDNA) reads. Trimmed reads were aligned to the assembled genome using HISAT2 version 2.1.0 [22,23] and sorted using PICARD version 2.4.0 [24]. Intron hints were generated from the aligned RNA sequences using the “bam2hints” utility provided by AUGUSTUS version 3.3 [25]. We tested values for the parameter maximum intron lengths (--maxintronlen) ranging from 10,000 to 1,000,000 bp in bam2hints and selected the value of 400,000 bp to generate the largest number of intron hints. REPEATMASKER version 4.0.7 [26] was used to mask interspersed repeats in the assembled genomes. We conducted gene prediction on the repeat-masked genomes based on the training files and intron hints using BRAKER1. 

### 2.5. Assessing Assembled Genome Quality

We estimated assembly statistics such as the numbers, lengths, and N50 values of scaffolds and the lengths of assembled genomes using QUAST version 4.5 [27]. We also estimated the numbers and lengths of gaps in the scaffolds using ASSEMBLY-STATS version 1.0.1 [28]. We calculated the computational times (i.e., running time x CPU number) of the genome assembly processes.

We used BUSCO to evaluate the assembly completeness of all assembled genomes based on 4915 AVES genes. In addition, we evaluated the completeness of predicted transcripts and genes (the BRAKER1 results) in the PLATANUS-assembled genomes based on the presence of start and stop codons. We defined a “complete transcript” as one that contained both a start and stop codon. Furthermore, we defined a “complete gene” as one with at least one “complete transcript”. Higher percentages of complete transcripts and genes indicate higher levels of genome assembly completeness.

We aligned PLATANUS-assembled genomes (excluding scaffolds shorter than 5000 bp) against three representative genomes (excluding unlocalized scaffolds) in birds (zebra finch: GCA_000151805.2, collared flycatcher: GCA_000247815.2, and red jungle fowl (chicken): GCA_000002315.5), respectively, using MINIMAP2 version 2.10 [29] with the default settings. The zebra finch genome and the collared flycatcher genome contained only the Z chromosome and the chicken genome contained both Z and W chromosomes. We generated dot plots to visualize the alignment results using D-GENIES [30]. We also generated circle diagrams (Jupiter plots) using JupiterPlot [31], where the assembled genomes were aligned to the reference genomes using BWA version 0.7.17 [32] with default settings and diagrams were plotted using CIRCOS version 0.69-6 [33].

### 2.6. Machine Specifications

All the analyses were run on an Ubuntu 16.04 server, with 48 threads on 24 cores (two 12 core processors: Intel Xeon processer E5-2690 version 3) and 775 GB RAM.

## 3. Results

### 3.1. Genome Assembly

We obtained 75.8 Gb paired-end reads with an average estimated fragment length of 436 ± 57 (SD) bp, and 126.7 Gb mate-pair reads with six average estimated fragment lengths, 1.96 ± 0.43, 4.11 ± 0.71, 6.50 ± 0.78, 9.19 ± 1.18, 12.29 ± 2.11, and 14.96 ± 4.36 Kb (the fragment lengths were estimated using PLATANUS). Based on the paired-end reads created from three trimming strategies (i.e., raw PE, trimmed PE, and cut off PE) and two assembly approaches, we generated six assembled genomes.

The genome assembly results based on PLATANUS were generally better than those based on DISCOVARdenovo + SOAPdenovo2 (Table 1 and Table 2). Although the total lengths of assembled genomes were similar between the two approaches (1.09–1.25 Gb), the scaffold N50 values of the PLATANUS-assembled genomes (13.31–16.89 Mb) were 7–10 times longer than those of the DISCOVARdenovo+SOAPdenovo2-assembled genomes (1.42–2.55 Mb; Table 1 and Table 2). The former contained fewer scaffolds (660–838 scaffolds > 5000 bp) than the latter (2617–3231 scaffolds > 5000 bp; Table 1 and Table 2). Although the number of gaps in the PLATANUS-assembled genomes (46,333–52,286) were slightly higher than those of the DISCOVARdenovo+SOAPdenovo2-assembled genomes (33,342–49,601), the unknown base (N) frequencies in the former (8.19–9.22 Ns per 1000 bp) were lower than those in the latter (28.64–39.62 Ns per 1000 bp) for all three datasets (Table 1 and Table 2). Based on the N50 values, scaffold numbers, and average N frequencies, we chose the PLATANUS-assembled genomes for downstream analyses (i.e., gene prediction) over the DISCOVARdenovo+SOAPdenovo2-assembled ones.

The computational times for the PLATANUS analyses were longer than those for the DISCOVARdenovo + SOAPdenovo2 analyses, especially for those based on the raw PE dataset (Table 3 and Table 4). The computational time for assembling the raw PE dataset using PLATANUS (14,272 CPU × hour) was around 15 times longer than that using DISCOVARdenovo + SOAPdenovo2 (951 CPU × hour); those for the trimmed PE and cut off PE datasets using the first approach (3640 and 2939 CPU × hour, respectively) were only about 2.5 times longer than those using the second one (1625 and 1113 CPU × hour, respectively; Table 3 and Table 4). The computational time for assembling the raw PE datasets was around 4–5 times longer than those for the trimmed PE and cut off PE datasets using PLATANUS (Table 3); in contrast, the difference in the computational times for assembling the three datasets using DISCOVARdenovo + SOAPdenovo2 were relatively small (Table 4).

### 3.2. Assembly Completeness

The BUSCO analyses showed that the PLATANUS-assembled and DISCOVARdenovo+SOAPdenovo2-assembled genomes contained similar levels of assembly completeness, regardless of which PE dataset was used (93.8–94.9%; Table 5 and Table 6). The predicted gene numbers of the three PLATANUS-assembled genomes were also similar and close to those of most published avian genomes (N ≅ 20,000), although the raw PE assembled genome had the highest number of genes (raw PE: 21,919 genes; trimmed PE: 21,712 genes; cut off PE: 20,859 genes; Table 7). The average length of the predicted coding genes for the raw PE assembled genome (17,910 bp) was slightly shorter than those of the other two assembled genomes (trimmed PE: 18,375 bp; cut off PE: 18,766 bp; Table 7). The average lengths of the exons of the three assembled genomes were similar (173, 171, and 169 bp for the raw PE, trimmed PE, and cut off PE, respectively), and so were those of the introns (2457, 2456, and 2465 bp, respectively; Table 7). According to the presence of a start and stop codon, the completeness levels of the predicted transcripts and genes of the trimmed PE genome (complete transcript: 94.0%; complete gene: 93.1%) were similar to those of the other two genomes (complete transcript: raw PE = 93.8% and cut off PE = 93.9%; complete gene: raw PE = 92.9% and cut off PE = 92.9%; Table 8).

### 3.3. Comparisons among the Rufous-Capped Babbler Genomes and Three Other Avian Genomes

The D-GENIES plots showed that assembled genomes based on different read datasets had somewhat different scaffold structures (Figure 1 and Appendix A). The PLATANUS-assembled genomes based on trimmed PE showed fewer cases of reversion than those of the other two datasets when compared to any of the three reference genomes (i.e., the zebra finch, collared flycatcher, and chicken genomes; Figure 1 and Appendix A). Interestingly, the alignment differences among different read datasets were generally more dramatic than those among the different reference genomes. Nevertheless, comparisons against the chicken genome still showed more cases of reversion than those against the zebra finch or collared flycatcher genomes. The Jupiter plots showed similar patterns: the raw PE genome had more split alignments when compared to those of the other two datasets (Figure 2 and Appendix A). 

We found that the assembled scaffolds generally corresponded well to the chromosomes of the reference genomes with a few long scaffolds in each chromosome (Figure 2 and Appendix A). However, the scaffolds corresponding to the Z chromosome were poorly assembled in all three datasets, regardless of the reference genome, and so were those corresponding to the W chromosome when aligned to the red jungle fowl genome (the other two reference genomes did not have the W chromosome; Figure 2 and Appendix A). In addition, scaffolds corresponding to chromosome 4A were also poorly assembled when aligned to the zebra finch and collared flycatcher genomes, and so were those corresponding to parts of chromosome 4 when aligned to the red jungle fowl genome (which did not have chromosome 4A, but did have a longer chromosome 4 than the two passerine genomes; Appendix A).

## 4. Discussion

Trimming low quality bases from sequence reads is considered not only as the first step to improve genome assembly quality, but also a possible way to determine the computational time [6]. As the cost of genomic sequencing decreases, one of the major concerns remaining for genomic projects is the amount of computational time needed to obtain reliable results. In this study, we examined how different trimming strategies affected the efficiency of assembling the rufous-capped babbler genome. 

Researchers often need to consider three factors when assembling a genome: (1) sequence read quality, (2) assembled genome quality, and (3) computational time for assembly. These three factors could be highly correlated. Lower quality reads may result in less accurate and less complete assembled genomes and longer computational times. Thus, removing low quality bases from sequence reads may seem to be the best strategy. However, our results showed an inconsistent relationship between the three factors when using different assembly approaches (i.e., PLATANUS and DISCOVARdenovo + SOAPdenovo2), except that the lengths of the scaffolds (i.e., N50) were always longer for lower quality, untrimmed reads (i.e., the raw PE dataset) than higher quality, trimmed ones (i.e., the trimmed PE and cut off PE datasets). Among the three PLATANUS-assembled genomes, the genome assembled from the trimmed PE dataset had a medium scaffold N50 value and took a medium amount of computational time, but yielded the fewest Ns and gaps (Table 1). In contrast, the trimmed PE genome had a medium scaffold N50 value and medium numbers of Ns and gaps, but its computation time was the longest of the three DISCOVAR+SOAPdenovo2 assembled genomes (Table 2).

The trade-offs between computational time and genome quality (e.g., genome contiguity and assembly completeness) should be considered based on the goals of the project and assembly approach. In this study, the two assembly approaches took relatively different amounts of time among the three datasets. However, given that the PLATANUS-assembled genomes yielded much (7–10 times) larger scaffold N50 values (i.e., better genome contiguity) than the DISCOVARdenovo+SOAPdenovo2 ones, we discuss the effect of trimming strategies on assembly quality based only on the former. A previous study suggests that the effect of read trimming on genome assembly might be detrimental as more aggressive trimming might result in lower scaffold N50 values and less accurate assemble rates, although they did not conduct gene prediction for the assembled genomes [6]. Surprisingly, we found few differences in the numbers and completeness of predicted genes and transcripts among the three PLATANUS-assembled genomes, indicating similar levels of assembly (gene prediction) completeness. That is, read-trimming had little impact on assembly completeness for coding genes.

We also found that the computational times increased 292%–383% while the N50 values only increased 8%–27% when assembling untrimmed reads compared with trimmed reads. Thus, if reference genome contiguity is not critical for focal studies such as those that aim to conduct SNP calling or RNAseq mapping, then one should at least perform a gentle trimming step to save a considerable amount of computational time, especially when using PLATANUS, as this does not appear to decrease assembly completeness. For studies focusing on genomic structural variation, one might consider assembling reference genomes based on untrimmed reads to increase genome contiguity; nevertheless, the computational time needed to assemble untrimmed reads might be too long and not affordable for every team.

Longer assembled scaffolds, however, do not always guarantee a more correct assembly. Although the PLATANUS-assembled genome based on the raw PE dataset with the longest scaffolds (scaffold N50 = 16.9 MB) revealed more cases of reversion than that based on the trimmed PE dataset (scaffold N50 = 15.6 MB), that based on the cut off PE dataset with the shortest scaffolds (scaffold N50 = 13.3 MB) also showed a higher reversion rate than that based on the trimmed PE dataset (Appendix A). The results imply that assembled genomes with longer scaffolds do not guarantee more accurate estimates of genome structure changes, especially if mis-assembly such as incorrect joining between mis-oriented segments occurs [34]. Raw reads might generate longer scaffolds with more mis-assemblies than trimmed reads because the former have higher levels of sequencing errors or adapter contamination than the latter. The read-trimming practice could reduce these error resources and thus lower the frequencies of mis-assemblies.

The scaffolds corresponding to the chromosome Z, W, and 4A were poorly assembled in all of the three read datasets. Interestingly, parts of chromosome 4A have been found translocated into the W and Z chromosomes, forming neo-sex chromosomes in Sylvioidea, an avian superfamily that includes babblers [35,36]. The complex genomic structure caused by the fusion history might make the neo-sex chromosomes difficult to assemble. In addition, a richness of repeats and the haploid nature of sex chromosomes may also increase assembly difficulties [37,38]. Overall, all of the PLATANUS-assembled genomes based on the three datasets with different trimming strategies performed similarly when being used to detect such genomic structure changes, while their computational times for assembly varied considerably.

## 5. Conclusions

Read trimming, which is thought be to a necessary step when analyzing NGS data, should be evaluated before being adopted as it might have unexpected drawbacks (e.g., [7]). Optimizing trimming strategies depends on the downstream applications of assembled genomes and the available computational time. Only by considering the trade-offs between different trimming strategies will researchers determine the best approaches for their own genome assembly projects. For example, if the rufous-capped babbler genome generated in this study was to be applied to identify genetic variations (SNPs) contributing to their adaptation to diverse climatic niches, we would have assembled the genome using PLATANUS based on the trimmed PE read dataset to reduce the computational time without cost to the assembly completeness; computational time is an important but often overlooked factor in genomic projects. In addition, the rufous-capped babbler genome with long scaffolds and high gene annotation quality presented in this study can provide a good system to study avian ecological adaptation in East Asia.

## Figures and Tables

**Figure 1 genes-10-00737-f001:**
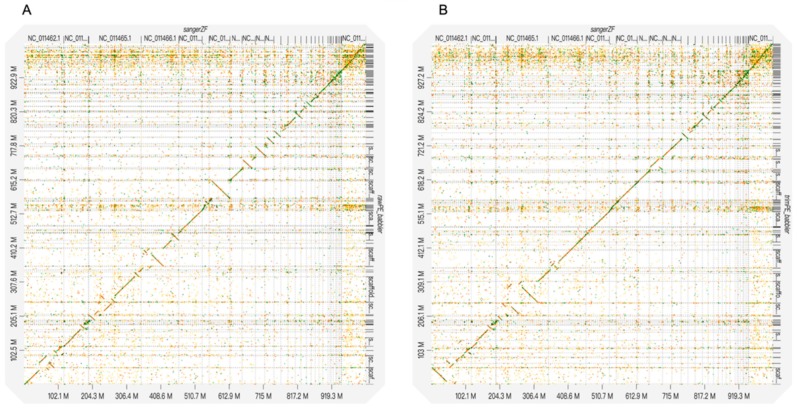
D-GENIES plots of the alignments between the rufous-capped babbler and zebra finch genomes. The raw PE (**A**) and trimmed PE (**B**) PLATANUS-assembled genomes are mapped to the zebra finch genome. Only the scaffolds of the rufous-capped babbler genomes with lengths > 5000 bp were used. The unlocalized scaffolds of the zebra finch genome were excluded from the analyses.

**Figure 2 genes-10-00737-f002:**
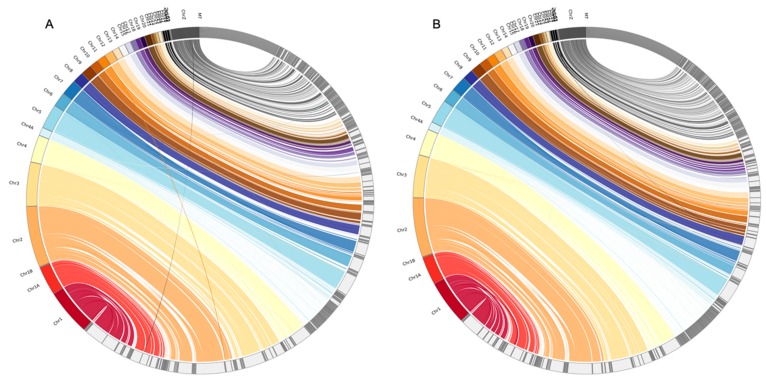
Jupiter plots of the alignments between the rufous-capped babbler and zebra finch genomes. The raw PE (**A**) and trimmed PE (**B**) PLATANUS-assembled genomes are mapped to the zebra finch genome. Only the scaffolds of the rufous-capped babbler genomes with lengths > 5000 bp were used. The unlocalized scaffolds of the zebra finch genome were excluded from the analyses.

**Table 1 genes-10-00737-t001:** De novo assembly results of rufous-capped babbler genomes based on three datasets using PLATANUS. The summaries are based on scaffolds ≥ 1000 bp, with exceptions noted in the rows.

PLATANUS	Raw PE	Trimmed PE	Cut Off PE
# scaffolds (>= 0 bp)	977,450	372,099	640,832
# scaffolds	9926	7756	8660
# scaffolds (>= 5000 bp)	660	742	838
Total length (>= 0 bp)	1,154,981,082	1,092,952,113	1,118,080,274
Total length	1,040,050,024	1,041,446,934	1,028,622,565
Total length (>= 5000 bp)	1,025,650,842	1,030,254,425	1,016,176,548
Largest scaffold	74,755,707	64,119,864	50,901,048
GC (%)	42.07	42.14	42.05
Scaffold N50	16,893,686	15,643,638	13,305,708
Gaps	48,535	46,333	52,286
N_count	9,597,849	8,524,882	9,456,690
N per 1000 bp	9.22	8.19	9.19

**Table 2 genes-10-00737-t002:** De novo assembly results of rufous-capped babbler genomes based on three datasets using DISCOVARdenovo + SOAPdenovo2. The summaries are based on scaffolds ≥ 1000 bp, with exceptions noted in the rows.

DIS + SOAP	Raw PE	Trimmed PE	Cut Off PE
# scaffolds (>= 0 bp)	392,770	244,234	326,322
# scaffolds	20,349	20,600	25,332
# scaffolds (>= 5000 bp)	2617	2942	3231
Total length (>= 0 bp)	1,239,771,083	1,212,202,269	1,250,242,340
Total length	1,131,554,057	1,138,451,258	1,151,285,864
Total length (>= 5000 bp)	1,100,826,487	1,107,888,121	1,114,172,265
Largest scaffold	14,426,870	14,721,577	10,390,167
GC (%)	42.44	42.42	42.39
Scaffold N50	2,5527,00	1,995,139	1,419,184
Gaps	33,342	40433	49601
N_count	32,412,091	39,082,416	45,610,276
N per 1000 bp	28.64	34.33	39.62

**Table 3 genes-10-00737-t003:** Computational times for genome assembly based on three datasets using PLATANUS. The computational times for three genome assembly procedures, contig assembly, scaffolding, and gap closing, were estimated, separately.

	Raw PE			Trimmed PE			Cut Off PE		
	Time (mins)	CPU	Time × CPU	Time (mins)	CPU	Time × CPU	Time (mins)	CPU	Time × CPU
Contig assembly	21,131	40	**845,240**	5008	40	**200,320**	3899	40	**155,960**
Scaffolding	622	1	**622**	507	1	**507**	500	1	**500**
Gap Closing	262	40	**10,480**	440	40	**17,600**	497	40	**19,880**
SUM (mins)			856,342			218,427			176,340
SUM (hours)			**14,272**			**3640**			**2939**

**Table 4 genes-10-00737-t004:** Computational times for genome assembly based on three datasets using DISCOVARdenovo + SOAPdenovo2. The computational times for three genome assembly procedures, contig assembly, scaffolding, and gap closing, were estimated, separately.

	Raw PE			Trimmed PE			Cut Off PE		
	Time (mins)	CPU	Time × CPU	Time (mins)	CPU	Time × CPU	Time (mins)	CPU	Time × CPU
Contig assembly	834	40	**33,360**	1803	40	**72,120**	1214	40	**48,560**
Scaffolding	205	40	**8200**	366	40	**14,640**	290	40	**11,600**
Gap Closing	387	40	**15,480**	224	48	**10,752**	138	48	**6624**
SUM (mins)			57,040			97,512			66,784
SUM (hours)			**951**			**1625**			**1113**

**Table 5 genes-10-00737-t005:** Assembly completeness of three PLATANUS-assembled genomes based on BUSCO analyses. Types of BUSCOs indicate the assessment output types of benchmarking universal single-copy orthologs.

	Raw PE		Trimmed PE		Cut Off PE	
Types of BUSCOs	*N*	%	*N*	%	*N*	%
Complete	4623	94	4642	94.5	4630	94.2
Complete and single-copy	4572	93	4595	93.5	4588	93.3
Complete and duplicated	51	1	47	1	42	0.9
Fragmented	172	3.5	155	3.2	159	3.2
Missing	120	2.5	118	2.3	126	2.6
Total	4915	100	4915	100	4915	100

**Table 6 genes-10-00737-t006:** Assembly completeness of three DISCOVARdenovo+SOAPdenovo2-assembled genomes based on BUSCO analyses. Types of BUSCOs indicate assessment output types of benchmarking universal single-copy orthologs.

	Raw PE		Trimmed PE		Cut Off PE	
Types of BUSCOs	*N*	%	*N*	%	*N*	%
Complete	4668	94.9	4663	94.9	4607	93.8
Complete and single-copy	4588	93.3	4575	93.1	4530	92.2
Complete and duplicated	80	1.6	88	1.8	77	1.6
Fragmented	146	3	160	3.3	180	3.7
Missing	101	2.1	92	1.8	128	2.5
Total	4915	100	4915	100	4915	100

**Table 7 genes-10-00737-t007:** BRAKER gene prediction results for the PLATANUS-assembled genomes.

	Raw PE			Trimmed PE			Cut Off PE		
	Number	Total length	Mean length	Number	Total length	Mean length	Number	Total length	Mean length
**Gene**	21,919	392,576,027	17,910	21,712	398,950,687	18,375	20,859	391,437,927	18,766
**Transcript**	25,736	586,294,213	22,781	25,594	594,536,470	23,230	24,651	586,027,670	23,773
**start_codon**	24,849	74,547	3	24,708	74,124	3	23,797	71,391	3
**stop_codon**	24,769	74,307	3	24,676	74,028	3	23,735	71,205	3
**exon**	245,952	42,525,218	173	249,072	42,562,276	171	244,448	41,219,021	169
**intron**	221,329	543,768,995	2,457	224,695	551,974,194	2,456	220,985	544,808,649	2,465

**Table 8 genes-10-00737-t008:** Completeness of predicted genes and transcripts in three PLATANUS-assembled genomes based on the presence of start and stop codons. Com_G indicates complete predicted genes. Com_T indicates complete predicted transcripts. %Com_G indicates the percentage of predicted genes is complete. %Com_T indicates the percentage of predicted transcripts is complete.

	Raw PE	Trimmed PE	Cut Off PE
Gene	21,919	21,712	20,859
Com_G	20,369	20,220	19,382
%Com_G	92.9%	93.1%	92.9%
Transcript	25,736	25,594	24,651
Com_T	24,142	24,051	23,135
%Com_T	93.8%	94.0%	93.9%

%Com_G = (Com_G / Gene) × 100%; %Com_T = (Com_T / Transcript) × 100%.

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
