# Peer review of "To Trim or Not to Trim: Effects of Read Trimming on the De Novo Genome Assembly of a Widespread East Asian Passerine, the Rufous-Capped Babbler (Cyanoderma ruficeps Blyth)"

_genes, 2019, doi:10.3390/genes10100737_

Round 1
Reviewer 1 Report
This article by “Yang and Lu et al” describes the effects of read trimming on the genome assemblies. Using Illumina reads from Asian passerine, the authors generated the genome assemblies using two different assemblers and three trimming strategies. Furthermore, all these assemblies were compared and evaluated at various factors to check the quality and the completeness of the final genomes.
First, of all, I would like to thank the authors for this study. In genome/transcript assemblies, often researchers found themselves in a dilemma of whether to trim the raw reads or not. Glad that now we have one more study that describes the advantages and disadvantages of read trimming.
The article is well written, the method is described sufficiently. Overall, the study is well conducted. I have a few minor comments/suggestions for the authors:
Lines 116-117: Scaffolds shorter than 1,000 bp removed by the authors from the downstream analysis, could encode for short hypothetical proteins? And this could be the potential reason for some of the missing genes in the assembled genome? Line 213-216: Total number (shorter length) of the predicted genes in raw PE assembly could be due to the presence of a higher number of split/fragmented genes? I wonder if authors observed genes which are split across multiple scaffolds? Table 7: To make it clearer for the reader, please define how the % of complete genes and transcripts were calculated in table footnote. For instance, “cut off PE” and “raw PE" assemblies both have 92.9% of complete genes even though the former has a lower number of genes. Did authors look for the genes in the unlocalized scaffolds as well? If not, then this could be another potential reason for missing genes in the assemblies. I believe if authors discuss the effect of trimming on the completeness of the genomes – in respect to fragmented and missing genes in the discussion section then this would be beneficial for the readers.
Author Response
Responses to Reviewers’ Comments
We thank the editor and reviewers for their time and effort in helping us improve our manuscript. Our answers to the reviewers’ comments are after the symbol “>>”.
This article by “Yang and Lu et al” describes the effects of read trimming on the genome assemblies. Using Illumina reads from Asian passerine, the authors generated the genome assemblies using two different assemblers and three trimming strategies. Furthermore, all these assemblies were compared and evaluated at various factors to check the quality and the completeness of the final genomes.
First, of all, I would like to thank the authors for this study. In genome/transcript assemblies, often researchers found themselves in a dilemma of whether to trim the raw reads or not. Glad that now we have one more study that describes the advantages and disadvantages of read trimming.
The article is well written, the method is described sufficiently. Overall, the study is well conducted. I have a few minor comments/suggestions for the authors:
Lines 116-117: Scaffolds shorter than 1,000 bp removed by the authors from the downstream analysis, could encode for short hypothetical proteins? And this could be the potential reason for some of the missing genes in the assembled genome?
>> It is possible that some of the removed scaffolds < 1,000 bp encode very short proteins. Even so, the impact should be little because the assembled genomes still contain around 21,000 predicted genes, similar to those of other published avian genomes. Even though BUSCO analyses showed missing genes rates of about 2.5% for the assembled genomes, the results might reflect technical limitations (e.g., the gene prediction step conducted by BUSCO) rather than the real missing gene rates (Simão et al. 2015, Bioinformatics). In fact, the BUSCO results are more suitable to assess relative assembly completeness among genomes than to estimate their real completeness levels.
Line 213-216: Total number (shorter length) of the predicted genes in raw PE assembly could be due to the presence of a higher number of split/fragmented genes? I wonder if authors observed genes which are split across multiple scaffolds?
>> The raw PE assembled genome has a slightly fewer complete predicted genes (92.9%) than that of the trimmed PE assembled genome (93.1%; see Table 8). We also observe some cases that two predicted genes in different scaffolds can be blasted to the same genes in the chicken genome. However, most of them are paralogous genes in the rufous-capped babbler genome (i.e., genes that duplicate in the rufous-capped babbler genome), rather than fragmented genes. In fact, the interpretation of the blast analytic results is not always straightforward. Thus, although the slightly shorter predicted genes in raw PE assembly can potentially be caused by more fragmented genes, the evidence we have is not strong. Also, the differences in gene prediction results among the assemblies are very small. Thus, we decide not to make exaggerated inference about this point.
Table 7: To make it clearer for the reader, please define how the % of complete genes and transcripts were calculated in table footnote. For instance, “cut off PE” and “raw PE" assemblies both have 92.9% of complete genes even though the former has a lower number of genes.
>> What we mean is that the percentage of the predicted genes or transcripts is complete. We recognize that the original abbreviated term (% of Com_G) is confusing, and thus we changed it to %Com_G. We also added the statement “%Com_G indicates percentage of the predicted genes is complete. %Com_T indicates the percentage of predicted transcripts is complete” in the table legend, and added the equations, %Com_G = (Com_G / Gene) × 100%; %Com_T = (Com_T / Transcript) × 100%, in the table footnote. We also changed the values of the two parameters to percentages, e.g., from 92.9 to 92.9%.
Did authors look for the genes in the unlocalized scaffolds as well? If not, then this could be another potential reason for missing genes in the assemblies. I believe if authors discuss the effect of trimming on the completeness of the genomes – in respect to fragmented and missing genes in the discussion section then this would be beneficial for the readers.
>> We think that the reviewer might misunderstand our analyses here. The unlocalized scaffolds are the scaffolds of the reference genomes (i.e., the zebra finch, collared flycatcher and chicken genomes) that cannot be assigned to a chromosome by their original authors. In fact, we did gene prediction for all of assembled scaffolds, regardless whether they were mapped to a chromosome or unlocalized scaffolds of the reference genomes. Thus, the predicted genes already included ones that can be mapped to the unlocalized scaffolds. That is, genes in unlocalized scaffolds cannot explain the missing or fragmented predicted genes. As we have discussed in previous comments, we do not think that we have many missing or fragmented genes in the assembled genomes. Also, our results showed that different trimming strategies make little difference in genome assembly (predicted gene) completeness, and we already discussed this point in the Discussion (Line 360-362). Now we add one more sentence to further stress this point (Line 262-263).
Reviewer 2 Report
The authors present an interesting study examining the effects of read trimming on genome assembly. The authors argue that read trimming may not yield the optimal assembly since raw read assemblies yield longer genome assemblies. Unfortunately, the authors have not fully evaluated the situations that could lead to these assemblies. Longer assemblies could in fact be more erroneous due to (1) extraneous contigs deriving from sequence errors and (2) misassemblies due to adapter contamination allowing false sequence joins. The manuscript needs to be significantly revised to examine these issues.
Additional specific comments:
Lines 77-78: What were the expected insert sizes for short and mate-pair libraries? I see that the recovered insert sizes are in the results, but it would be ideal to know what the targets were in the methods section.
Lines 92-95: Are these additional species currently being studied in your laboratory. What are the scientific binomial names for the species in the default and addended Kraken databases?
Line 106 (and elsewhere): "Genome assembling" should be "Genome assembly"
Lines 113-114. Rephrase: We tested a series of kmer lengths ranging from 25 to 121 bp, and selected the 25-mer that generated scaffolds with the longest N50 value.
Currently indicates that the authors chose a specific 25-mer, rather than assembled using 25mers.
Lines 129-130: Given the focus of this manuscript on the effects of trimming, it might be worthwhile investigating the effects of trimming on gene prediction as well. The authors could repeat the analysis using the untrimmed RNA reads and see if results are similar or different.
Line 162: "48-thread (CPU)": This phrase is confusing and inaccurate. I presume the authors mean that they ran it using 48 threads on 24 cores (two 12 core E5-2690 version 3: Intel processor specifications) since there is no 24-core processor in this processor family.
Lines 125 and Results: It would be ideal to run BUSCO on the DISCOVAR+SOAPdenovo2 libraries as well. This would provide another source of data that the PLATANUS libraries are indeed better, rather than simply more contiguous due to erroneous joins.
Tables 1 and 2: The trimmed/cut off reads had significantly fewer contigs with a very similar genome size. This suggests to me that the extra contigs in the less trimmed datasets (the cut off set and especially the raw set) represent possible misassemblies from sequence errors. These misassemblies would then add to the overall sequence length.
Lines 236-245: Couldn't these extra reversions be explained by misassemblies due to adapter contamination?
Author Response
Responses to Reviewers’ Comments
We thank the editor and reviewers for their time and effort in helping us improve our manuscript. Our answers to the reviewers’ comments are after the symbol “>>”.
Comments and Suggestions for Authors
The authors present an interesting study examining the effects of read trimming on genome assembly. The authors argue that read trimming may not yield the optimal assembly since raw read assemblies yield longer genome assemblies. Unfortunately, the authors have not fully evaluated the situations that could lead to these assemblies. Longer assemblies could in fact be more erroneous due to (1) extraneous contigs deriving from sequence errors and (2) misassemblies due to adapter contamination allowing false sequence joins. The manuscript needs to be significantly revised to examine these issues.
>> We agree that sequencing errors and adapter contamination in raw reads could be main reasons of mis-assembly. Our results based on comparisons among raw and trimmed reads (in which sequencing errors and contaminated adapters were reduced) demonstrate the possible effects of these error resources on genome assembly. Thus, we added discussion in the issues in the revised manuscript (Lines 380-383).
However, separating the effects of the two resources on genome assembly is not trivial and most researchers would not choose to remove one of them if they can do both in one analysis. In addition, FastQC reported a low level of adaptor contamination in our sequencing reads (< 3.5% of reads potentially contain any contaminated adapter sequences near their tail ends; lower than the warning threshold of FastQC). Trimmomatic’s original paper (Bolger et al. 2014) also suggests relatively little impact of adaptor contamination on genomic analyses compared with low sequencing quality. Furthermore, the main focus of this study is to examine how trimming practices affect the efficiency of genome assembly, rather than to separate the possible error resources in sequencing reads. Thus, we did not conduct extra analyses on this part that may make the manuscript unnecessarily long and complex.
Additional specific comments:
Lines 77-78: What were the expected insert sizes for short and mate-pair libraries? I see that the recovered insert sizes are in the results, but it would be ideal to know what the targets were in the methods section.
>> What we presented in the Results section are expected fragment (insert) sizes, not recovered ones. To reduce confusion, we moved the expected fragment size data to the Methods section in the revised manuscript (Lines 88-89). We also added the average estimated insert sizes with standard deviations of the pair-ended and mate pair libraries in the Results sections in the revised manuscript (Lines 188-191).
Lines 92-95: Are these additional species currently being studied in your laboratory. What are the scientific binomial names for the species in the default and addended Kraken databases?
>> We added the 7 taxa (species) because our laboratory and neighbor ones in our buildings study them and we share some experimental benches. Thus, it is possible that their samples could contaminate ours. We added the scientific names for the 7 species in the manuscript (Lines 106-108). Each taxon in the default Kraken database (human, archaea, bacteria, plasmid, and viral) contain many species except for the human one, and thus we cannot list all of their scientific names in the manuscript, and they can be found in the Kraken manual and website.
Line 106 (and elsewhere): "Genome assembling" should be "Genome assembly"
>> We changed “assembling” to “assembly” here and in Lines 226, 389 and 392.
Lines 113-114. Rephrase: We tested a series of kmer lengths ranging from 25 to 121 bp, and selected the 25-mer that generated scaffolds with the longest N50 value.
Currently indicates that the authors chose a specific 25-mer, rather than assembled using 25mers.
>> SOAPdenovo2 was used to perform scaffolding, one step for genome assembly. To make the statement clearer, we added “for scaffolding” in this sentence and rewrote parts of it. Now it becomes “We tested a series of kmer lengths ranging from 25 to 121 bp for scaffolding, and selected the 25-mer in this step to generate scaffolds with the longest N50 value.”
Lines 129-130: Given the focus of this manuscript on the effects of trimming, it might be worthwhile investigating the effects of trimming on gene prediction as well. The authors could repeat the analysis using the untrimmed RNA reads and see if results are similar or different.
>> We appreciate this comment, but testing the effect of RNA read-trimming will make the parameter space extremely large. If we try 3 different trimming strategies for RNA reads combined with 3 ones for paired-end DNA reads and two assembly approaches, there will be 3*3*2 = 18 combinations. It will make the results difficult to interpret and readers could be confused about what they can learn from this manuscript. That is why we only test trimming strategies in PE reads in this manuscript. In addition, RNA reads is used for gene prediction that is one step independent from genome assembly, and thus doing so is out of the scope of this manuscript.
Line 162: "48-thread (CPU)": This phrase is confusing and inaccurate. I presume the authors mean that they ran it using 48 threads on 24 cores (two 12 core E5-2690 version 3: Intel processor specifications) since there is no 24-core processor in this processor family.
>> We appreciate that the reviewer points out the unclear statement. We rewrote the sentence as “All the analyses were run on a Ubuntu 16.04 server, with 48 threads on 24 cords (two 12 core processers: Intel Xeon processer E5-2690 version 3) and 775 GB RAM.”
Lines 125 and Results: It would be ideal to run BUSCO on the DISCOVAR+SOAPdenovo2 libraries as well. This would provide another source of data that the PLATANUS libraries are indeed better, rather than simply more contiguous due to erroneous joins.
>> We follow the reviewer’s suggestion to add BUSCO analyses for the DISCOVARdenovo+SOAPdenovo2-assembled genomes. The results showed that assembly completeness levels were similar among the PLATANUS-assembled and DISCOVARdenovo+SOAPdenovo2-assembled genomes, regardless of which PE dataset was used (Lines 248-250).
Tables 1 and 2: The trimmed/cut off reads had significantly fewer contigs with a very similar genome size. This suggests to me that the extra contigs in the less trimmed datasets (the cut off set and especially the raw set) represent possible misassemblies from sequence errors. These misassemblies would then add to the overall sequence length.
>> We assume that the reviewer talked about scaffolds rather than contigs because we only showed the information of the former, not the latter, in Tables 1 and 2. We showed the numbers of all scaffolds, scaffolds longer than 1,000 bp, and scaffolds longer than 5,000 bp in the two tables. In fact, the raw PE genomes have more scaffolds, but fewer long scaffolds (> 5,000 bp), than the trimmed PE genomes. That is, the raw PE genomes tend to have more short scaffolds than the trimmed PE. Even so, we could not conclude that the raw PE genomes have more mis-assemblies than trimmed PE genomes based only on the above results because short scaffold can also be correct ones. However, more cases of reversions in the raw PE genomes than the trimmed PE genomes when compared to avian reference gnomes could somewhat suggest that the former may have more mis-assemblies than the latter. We already discussed this point in the Discussion (Lines 374 – 380) in the original manuscript.
Lines 236-245: Couldn't these extra reversions be explained by misassemblies due to adapter contamination?
>> As we discuss in the last comment, the more cases of revision in the raw PE genomes than the trimmed PE genomes might suggest more mis-assemblies in the former than the latter. The reason could be sequencing errors or adapter contamination, or both. In the revised manuscript, we added the possible reasons of mis-assembly in the Discussion (Lines 380-383).
Round 2
Reviewer 2 Report
The authors have addressed my previous comments in the current revision.
I only have two minor improvements:
On lines 102-103, I suggest that the authors change 'those of seven taxa lizard:' to 'those of seven common laboratory model taxa: lizard...' to clarify why these species need to be specifically excluded as contaminants.
On line 178, 'core' has been misspelled as 'cord'.